# Persistent Vitamin D Deficiency in Pediatric Patients with Cystic Fibrosis

**DOI:** 10.3390/nu17111890

**Published:** 2025-05-31

**Authors:** Magali Reyes-Apodaca, José L. Lezana-Fernández, Rodrigo Vázquez Frias, Mario E. Rendón-Macías, Aline González-Molina, Benjamín A. Rodríguez Espino, Isela Núñez-Barrera, Mara Medeiros

**Affiliations:** 1Faculty of Medicine, National Autonomous University of Mexico, 3004 Universidad Ave, Mexico City 04510, Mexico; maga.reyes@comunidad.unam.mx; 2Hospital Infantil de México Federico Gómez, 162 Dr. Marquéz Street, Mexico City 06720, Mexico; lezana_doc@yahoo.com.mx (J.L.L.-F.); rovaf@yahoo.com (R.V.F.); qfbalinehim@gmail.com (A.G.-M.); rodriguezespinoba@gmail.com (B.A.R.E.); nb.isela@gmail.com (I.N.-B.); 3Health Science Faculty, Medical School, Panamerican University, Donatello 85, Mixcoac, Mexico City 03920, Mexico; mrendon@up.edu.mx; 4Pharmacology Department, National Autonomous University of Mexico, 3004 Universidad Ave, Mexico City 04510, Mexico

**Keywords:** cystic fibrosis, vitamin D deficiency, vitamin D supplements

## Abstract

**Background/Objectives:** Cystic fibrosis (CF) is a multisystem disease caused by CFTR gene variants, with a high prevalence of vitamin D (VitD) deficiency despite the supplementation and schedules specifically developed for this population. Lower VitD levels have been associated with an increased risk of respiratory infections and pulmonary exacerbations in CF, with some pilot studies indicating the potential benefits of supplementation during acute episodes. This study aimed to describe the occurrence of VitD deficiency according to the supplemented dose in pediatric patients with CF. **Methods**: A cross-sectional analytical study was conducted to assess serum VitD levels in a pediatric population with cystic fibrosis. Clinical and biochemical data were collected, along with information on VitD intake and pancreatic enzyme dosage at the time of evaluation. **Results**: A total of 48 patients were included in the study. Normal VitD levels were observed in 41.7% of the patients, insufficiency in 31.3%, and deficiency in 27%. The median VitD intake was 2050 IU. A statistically significant difference was observed in patients with a daily intake exceeding 2000 IU. Only 10% of patients achieved levels above 30 ng/mL with a lower dose. No statistically significant association was identified between the pancreatic enzyme dosage and vitamin D levels. **Conclusions:** Vitamin D deficiency/insufficiency is a persistent problem in the pediatric CF population; the interventions targeting factors associated with this condition are required to refine supplementation schedules. These findings underscore the need for personalized strategies to optimize vitamin D status in PwCF. Ideally, these strategies should consider all associated factors, including genetic variants; however, with limited resources, our results suggest that a daily dose of 2000 IU of vitamin D may represent a reasonable and effective starting point for supplementation.

## 1. Introduction

Cystic fibrosis (CF) is an autosomal recessive disease caused by cystic fibrosis transmembrane conductance gene (CFTR) variants. The clinical manifestations of this condition are mainly due to a dysregulation of the balance between sodium absorption and chloride and bicarbonate secretion. This gene is expressed in the epithelial cells of the lung, salivary glands, sweat glands, pancreas, intestine, bile ducts, and kidney, resulting in a multisystemic disease [1,2].

Vitamin D (VitD) deficiency is common in CF patients (PwCF), particularly in those with pancreatic insufficiency. It has been linked to impaired pulmonary function, increased susceptibility to respiratory infections, and increased airway inflammation. Several studies have shown that higher serum vitamin D levels are associated with improvements in forced expiratory volume in one second (FEV1) values, as well as reduced rates of pulmonary exacerbations. Over 40% of these patients have values below 30 ng/mL and, in some reports, up to 80%, despite the different supplementation guidelines and even water-soluble vitamin supplements designed specifically for this population. There are differences between the guidelines for European and American CF patients regarding the cutoff value for initiating VitD supplementation (20 ng/mL vs. 30 ng/mL), as well as the dosing strategies and monitoring intervals. The high proportion of patients with insufficient VitD levels highlights important areas for improvement in care [3,4].

In the Mexican population, it has been reported that only 26.8% of preschoolers and 28% of school-aged children have sufficient vitamin D levels. However, to date, no studies have evaluated the vitamin D status in pediatric patients with cystic fibrosis in Mexico [5].

This study aims to determine the proportion of pediatric patients with cystic fibrosis attending the Cystic Fibrosis Center at the Hospital Infantil de México Federico Gómez who present with deficient or insufficient vitamin D levels in relation to the prescribed supplementation.

## 2. Materials and Methods

A cross-sectional analytical study was conducted on patients with cystic fibrosis diagnosed by molecular testing or sweat electrolyte analysis. The study protocol was approved by research and ethics committees (HIM-087). After informed consent was obtained, serum samples were collected to measure the 25-hydroxy-vitamin D levels using chemiluminescence with the LIAISON^®^ system (Diasorin, Saluggia, Italy). Blood samples for vitamin D analysis were collected in the early morning following a minimum 8 h fasting period. Vitamin D sufficiency was defined as levels > 30 ng/mL, insufficiency was 20–29.9 ng/mL, and deficiency was <20 ng/mL. Hypercalciuria and the urinary calcium/creatinine ratio were assessed through photometric analysis using the cobas c111 system (Roche Diagnostics Ltd.^®^, Burgess Hill, UK).

Demographic data were recorded, including age, sex, time since cystic fibrosis diagnosis, and treatment duration. The anthropometric assessment included weight, height, and body mass index (BMI) calculation. Nutritional indicators were analyzed according to age and sex, using reference parameters from the Centers for Disease Control and Prevention (CDC). Height was assessed using z-scores, with normality defined as ±2 standard deviations (SD) for age. Undernutrition was defined as a BMI < −2 SD, while severe malnutrition undernutrition was defined as a BMI < −3 SD. Based on the guidelines by Turck et al., a nutritional target for cystic fibrosis was set at a BMI above the 50th percentile [6].

Data on the prescribed vitamin supplements were collected, including the brand, form, content, presence of cholecalciferol (D3) or ergocalciferol (D2), dosage and time of supplement intake. Information on pancreatic enzyme supplementation (CREON^®^, Hertfordshire, UK) was also recorded, including dosage and formulation when applicable. All patients with exocrine pancreatic insufficiency were diagnosed at the time of admission to the center through a fecal pancreatic elastase test. Therapeutic adherence data were collected from the patient’s file, based on the regular clinical interviews; no other indirect methodologies were used to evaluate adherence.

For statistical analysis, the normality of quantitative variables was assessed using the Shapiro–Wilk test. Parametric variables are summarized as means and SD, while non-parametric variables are expressed as medians and interquartile ranges. Qualitative variables were analyzed using absolute frequencies and proportions. The χ^2^ test was used to determine the associations between variables based on the VitD classification. Statistical significance was set at *p* < 0.05. Due the prevalence of the CF in Mexico, we opted for a complete sampling approach of our center to ensure representation of the available population, therefore all patients treated in our center were invited to participate.

## 3. Results

From a total of 50 patients treated in our center, 48 agreed to participate, with a higher proportion of females (56.3%). The median age was 101.5 months. Exocrine pancreatic insufficiency was present in 89.6% of patients, with a median pancreatic enzyme supplementation dose of 5882 IU/kg/day (Table 1).

The nutritional status was determined based on BMI; 85.4% of patients were within the normal range, while 10.4% had undernutrition and 4.2% had severe undernutrition. Only 29.2% of patients met the recommended BMI target. In terms of linear growth, 15 patients (31.3%) were classified as having short stature (Table 1).

Serum VitD levels showed a median of 26.3 ng/mL. By classifying the levels, 41.7% of patients had sufficient levels, 31.1% had insufficiency, and 27% had a deficiency. At the time of evaluation, 83.3% of patients were receiving vitamin supplementation through a multivitamin containing varying doses of cholecalciferol or ergocalciferol, with a median intake of 2050 IU per day (Table 1).

A total of 40 patients received vitamin D supplementation, either in the form of cholecalciferol tablets or ergocalciferol contained within a multivitamin formulation. Among them, 11 patients received only multivitamins, while 29 were supplemented with a combination of multivitamins and cholecalciferol tablets. Only four patients (8.3%) had access to vitamin supplements specifically formulated for individuals with cystic fibrosis (Dekas^®^, Zeist, The Netherlands). Eight patients did not receive any form of vitamin D supplementation during the study period.

Patients were classified according to the reported VitD intake, comparing those with an intake greater than 2000 IU/day with those with an intake less than 2000 IU. A statistically significant difference was found in achieving vitamin D levels above 30 ng/mL for those ingesting more than 2000 IU daily (Figure 1).

No statistically significant association was found between those with sufficiency, insufficiency, or deficiency in the relationship between pancreatic enzyme dosage and VitD levels. A statistically significant association was identified between those ingesting more than 2000 IU per day and achieving sufficiency when analyzing vitamin D intake recommendations according to VitD levels (Table 2).

## 4. Discussion

Vitamin D deficiency remains a persistent challenge in the chronic management of patients with cystic fibrosis (CF). Various studies have reported prevalence rates ranging from 40% to 90% despite routine supplementation. These findings are consistent with the results of this study, in which 58% of our patients had vitamin D levels below 30 ng/mL, even though more than 80% were receiving vitamin D supplementation at the time of analysis. This persistent deficiency may be explained by multiple factors commonly observed in PwCF, including fat malabsorption due to exocrine pancreatic insufficiency, suboptimal adherence to supplementation protocols, and genetic variations affecting vitamin D metabolism and transport. In fact, it is estimated that approximately 11% of patients with CF may be classified as “non-responders” to supplementation due to such genetic variants. These factors may partially account for why some patients in our study, despite receiving high doses of vitamin D, did not achieve sufficient serum levels [7,8].

The role of vitamin D in bone health is well established, particularly in the regulation of calcium and phosphorous metabolism. In a systematic review by Farahbakhsh et al., the prevalence of vitamin D insufficiency (20–30 ng/mL) was 36%, and vitamin D deficiency (<20 ng/mL) was 27% in pediatric and adolescent patients. In contrast, prevalence rates of 63% and 35% have been reported in other studies. The prevalence rates observed in our study are consistent with these findings, with 27% of patients presenting with vitamin D deficiency and 31.3% showing insufficiency [9].

It has been reported that 20–35% of the adult CF population has reduced bone mineral density, whereas in pediatric patients, this prevalence can reach up to 38%. Adolescents appear to be at the highest risk, regardless of whether it is associated with pulmonary decline [3,4,10].

Exocrine pancreatic insufficiency leads to the malabsorption of nutrients, primarily fat and fat-soluble vitamins. To address this issue, specialized formulations, such as DEKAs^®^, have been developed for PwCF because they are water-soluble and can enhance absorption. In this study, only four patients received these specialized supplements, mainly because their acquisition depended on the caregivers [11].

Considering the relationship between malabsorption and fat-soluble vitamin deficiency, we analyzed daily pancreatic enzyme doses and vitamin D status. Contrary to our expectations, no statistically significant associations were observed. The lack of association in our study between pancreatic enzyme dose and vitamin D status may be multifactorial. Although enzyme therapy is essential to promoting vitamin absorption, our study did not control the timing of vitamin D intake in relation to enzyme administration or the frequency of enzyme intake. In routine clinical practice and considering that CF-specific vitamin formulations are not available in Mexico, patients are advised to take vitamin supplements with meals and pancreatic enzymes to optimize absorption.

However, the current recommendations support taking vitamin supplements alongside pancreatic enzymes, regardless of enzyme dose. The variability in adherence to this recommendation may have influenced our results. In addition, interindividual differences in the efficiency of gastrointestinal absorption may also play a role. These factors, although not directly measured, merit consideration in future studies aimed at optimizing fat-soluble vitamin supplementation in PwCF [7].

Currently, different vitamin D supplementation schedules have been suggested for CF. For example, the Cystic Fibrosis Foundation (CFF) aims for serum vitamin D levels >30 ng/mL, with dosage recommendations based on age and vitamin levels. The routine recommended dose for patients older than one year is 800–1000 IU/day for individuals aged up to 10 years, and 800–2000 IU/day for individuals over 10 years and adults.

For therapeutic doses, adjustments are made according to serum vitamin D levels, with a maximum of 4000 IU/day for patients aged 1–10 years, and up to 10,000 IU/day for those over 10 years and adults. In this study, patients receiving less than 2000 IU of vitamin D did not achieve sufficiency levels, suggesting that the CFF guidelines could serve as a useful reference for maintaining optimal vitamin D levels in the CF population. However, at least 16 patients in our study took 4000 IU or more at the time of assessment. Notably, three patients with vitamin D deficiency consumed more than 4000 IU/day, highlighting the need for further investigation [12,13].

Due to budget limitations for regular vitamin D testing, clinicians in resource-limited settings may face challenges in monitoring and individualizing care. In such settings, it is recommended that clinicians focus on clinical indicators of vitamin D deficiency, such as growth, bone health, and immune function, to guide supplementation. Additionally, since CF patients may face difficulties in accessing specialized formulations, alternative strategies to optimize supplementation based on clinical judgment, adherence, and available resources could be considered. While regular monitoring is ideal, in practice, some degree of individualized care may be necessary to ensure adequate vitamin D levels.

Our findings suggest that a daily intake of at least 2000 IU of vitamin D may be necessary to maintain sufficient serum levels in pediatric PwCF. This has important implications for local supplementation strategies, particularly in resource-limited settings where periodic serum monitoring may not be feasible. Our clinical team implemented a pragmatic approach based on these findings, prescribing 2000 IU/day for patients with vitamin D levels between 20 and 30 ng/mL, and 4000 IU/day for those with levels below 20 ng/mL. These results support the consideration of higher initial or maintenance doses, tailored to patient-specific factors and local resource availability.

Another set of guidelines for CF and vitamin D intake is the European Cystic Fibrosis Bone Mineralization Guidelines (ECFBMG), which recommend a target serum level of >20 ng/mL, in contrast to the 30 ng/mL recommended by the CFF. The ECFBMG does not define specific cut-off values for deficiency or insufficiency, but it generally recommends an initial dose of 1000–2000 IU/day for children under one year and 1000–5000 IU/day for older patients. The guidelines suggest increasing the dose based on serum levels; however, like the CFF guidelines, they do not specify monitoring intervals or how to adjust the doses over time. If we use the ECFBMG cut-off value, the prevalence of vitamin D deficiency remains significant [14].

Additionally, a study by Juhász, M. F. et al. [7] suggested that increasing vitamin D intake, usually between 1600 and 5000 IU, does not provide additional clinical benefits and may be unnecessary. However, it could help maintain sufficient vitamin D levels, even though the optimal duration for such an increase has not yet been specified. In our study, patients who achieved levels >30 ng/mL were administered more than 4000 IU/day. However, we did not consider the duration of this prescription or other factors such as sun exposure time, which may have influenced the results.

One of the limitations of our study is that we did not assess other clinical outcomes associated with vitamin D levels, such as pulmonary function and inflammatory markers. Other factors that were not evaluated include the presence of genetic variants in vitamin D receptors, which could affect its metabolism or transport. Beyond its role in bone metabolism, vitamin D plays an important role in modulating immune responses and inflammation—factors that are highly relevant in the context of CF. Previous studies have associated sufficient vitamin D levels with a reduced frequency of pulmonary exacerbations, fewer positive respiratory cultures, and lower levels of inflammatory markers. Therefore, the high prevalence of deficiency observed in our cohort may have broader clinical consequences, emphasizing the importance of adequate supplementation and monitoring [15,16].

Vitamin D regulation also depends on various physiological processes controlled by the vitamin D receptor (VDR), which is present in different tissues, including the bone, heart, and kidneys. The VDR can regulate the expression of approximately 500 genes and is activated upon binding to 1α,25(OH)_2_D, forming a heterodimer with the retinoid X receptor (RXR). The 1α,25(OH)_2_D-VDR-RXR complex then migrates to the nucleus, where it regulates the transcription of genes involved in vitamin D effects, including phosphorus and calcium metabolism, cell proliferation, and the regulation of innate and adaptive immunity [17].

In the meta-analysis conducted by Usategui et al., the most extensively studied allelic variants of the *VDR* gene—*Apal* (rs7975232), *Bsml* (rs1544410), *Taql* (rs731236), and *Fokl* (rs10735810)—were reviewed. The study demonstrated that the *Fokl* and *Taql* variants were associated with better responses to vitamin D supplementation, improved calcium absorption, higher bone mineral density, and a lower risk of fractures. In contrast, *Bsml* and *Apal* did not show significant associations [17].

It has been estimated that approximately 11% of CF patients can be classified as “non-responders”, due to the presence of genetic variants involved in vitamin D transport and metabolism, as they do not exhibit improved levels following standardized supplementation [17,18,19]. At least four variants have been identified as being associated with the supplementation response, which include the following: rs1352846, located in the intronic region of the *GC* gene, which encodes the vitamin D-binding protein (DBP) responsible for binding vitamin D and its metabolites for transport to target organs; and rs1800588, located in the intronic region of *LIPC* (hepatic lipase C). Additionally, two other SNPs—rs17216707 in the *CYP24A1* gene and rs116970203 in the *PDE3B* gene—were also found to be statistically significant in this association [19].

This study has several limitations. The cross-sectional design precludes causal inferences and does not allow for the assessment of temporal changes in vitamin D status. Although we employed a complete sampling approach of the pediatric CF population at our center to maximize representativeness, the single-center nature and relatively small sample size may limit external validity. Adherence to vitamin D supplementation and enzyme therapy replacement was not formally assessed, and factors known to influence vitamin D levels—such as sun exposure, seasonal variation, and dietary intake—were not documented. Clinical outcomes potentially related to vitamin D status, including pulmonary function and inflammatory markers, were also not evaluated. Lastly, genetic testing was not performed, which may have contributed to explaining the interindividual variability in response to supplementation. These limitations highlight the need for longitudinal, multicenter studies incorporating clinical, nutritional, behavioral, and genetic factors to optimize vitamin D management in this population.

Future research should address key areas to enhance our understanding and management of vitamin D deficiency in CF. Longitudinal studies are needed to evaluate adherence to vitamin D supplementation and its impact on serum levels over time. Additionally, controlled trials comparing supplementation schedules, dosing strategies (both initial and maintenance), and the duration required to sustain normal vitamin D levels could clarify their relative efficacy. Finally, genetic studies examining polymorphisms in vitamin D receptors and transport proteins may help identify individuals at risk for poor response to conventional supplementation, ultimately supporting more personalized approaches to therapy.

## 5. Conclusions

Vitamin D deficiency in PwCF is a prevalent condition that presents a therapeutic challenge, as it is involved in processes related to growth, inflammation, and immunity. While standardized supplementation and monitoring protocols remain insufficiently defined, our findings suggest that a daily intake above 2000 IU may be more effective in achieving sufficient serum levels. This highlights the need to refine current supplementation strategies, particularly in the Mexican pediatric CF population, to ensure more consistent and effective management of vitamin D deficiency.

## Figures and Tables

**Figure 1 nutrients-17-01890-f001:**
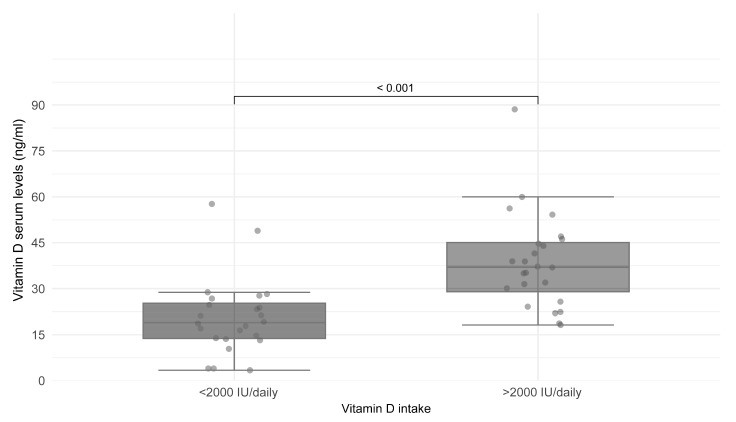
Comparison of vitamin D intake with serum levels in PwCF. Comparison of serum vitamin D levels (ng/mL) according to daily vitamin D intake. Serum 25-hydroxyvitamin D concentrations were significantly higher in participants receiving more than 2000 IU of vitamin D daily (median intake) compared to those receiving less than 2000 IU per day (*p* < 0.001). Data are presented as box plots showing median, interquartile range, and individual data points.

**Table 1 nutrients-17-01890-t001:** Clinical and demographic characteristics of subjects.

Pediatric Patients with Cystic Fibrosis	N = 48 Median [IQR]
Women, n (%)	27 (56.3)
Age, months	101.5 [68.5–140]
Exocrine Pancreatic Insufficiency, n (%)	43 (89.6)
PERT dose, IU/kg/day	5882.3 [4811–7272]
BMI Nutritional Status	
Normal, n (%)	41 (85.4)
Undernutrition, n (%)	5 (10.4)
Severe undernutrition n (%)	2 (4.2)
Nutritional BMI goal achieved, n (%)	14 (29.2)
Short stature, n (%)	15 (31.3)
OH-25-Hidroxi-Vitamin D, ng/mL	26.3 [18.3–38.9]
Vitamin D Status	
Sufficiency, n (%)	20 (41.7)
Insufficiency, n (%)	15 (31.3)
Deficiency, n (%)	13 (27)
Calcium/creatinine ratio	0.08 [0.027–0.158]
Use of vitamin D supplements, n (%)	40 (83.3)
Duration of supplement use, months	4 [3–6.75]
Vitamin D daily intake, IU	2050 [250–4350]

PERT: pancreatic enzyme replacement therapy, BMI: body mass index. Descriptive statistics include frequencies, median, and proportions.

**Table 2 nutrients-17-01890-t002:** Association between vitamin D levels, pancreatic enzyme dosages, and vitamin D intake.

	Sufficiency N = 20	Insufficiency N = 15	Deficiency N = 13	*p* Value
PERT doses, n (%)				
<5000 IU/kg/day	4 (20)	7 (46.6)	5 (38.5)	0.208 *
>5000 IU/kg/day	16 (80)	8 (53.4)	8 (61.5)	
Vitamin D intake, n (%)				
<2000 IU/day	2 (10)	13 (86.6)	9 (69.2)	0.001 *
>2000 IU/day	18 (90)	2 (13.4)	4 (30.8)	

PERT: pancreatic enzyme replacement therapy. * Chi-squared test comparing sufficiency/insufficiency vs. deficiency.

## Data Availability

The original contributions presented in this study are included in the article. Further inquiries can be directed to the corresponding author.

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
