# Peer review of "Persistent Vitamin D Deficiency in Pediatric Patients with Cystic Fibrosis"

_nutrients, 2025, doi:10.3390/nu17111890_

Round 1

Reviewer 1 Report

Comments and Suggestions for Authors

Abstract:

  • Area for Improvement: While the abstract clearly states the aim and main findings, it could benefit from briefly mentioning the implications of the persistent vitamin D deficiency. For example, a sentence like "These findings underscore the need for personalized strategies to optimize vitamin D status in this vulnerable population" could be added.
  • Limit: The abstract accurately reflects the cross-sectional nature of the study.

Introduction:

  • Area for Improvement: The introduction provides adequate background on CF and vitamin D deficiency. However, it could be strengthened by:
    • Quantifying the impact of vitamin D deficiency in CF: Briefly mentioning the specific consequences of vitamin D deficiency in CF patients (e.g., bone health, inflammation, pulmonary function) would emphasize the importance of the research question.
    • Elaborating on existing supplementation guidelines: Briefly mentioning the challenges or inconsistencies in current guidelines could further justify the study's aim to understand the local context.
  • Limit: The introduction appropriately highlights the lack of data in the Mexican pediatric CF population.

Materials and Methods:

  • Area for Improvement:
    • Sample Size Justification: While 48 patients provide a snapshot, a brief justification of the sample size in relation to the prevalence of CF in the region or feasibility constraints would be beneficial.
    • Vitamin D Supplementation Details: While the type (D2 or D3) is mentioned, providing more detail on the form of supplementation (e.g., liquid, tablets, within multivitamins) could be relevant as absorption might differ.
    • Adherence to Supplementation: The manuscript acknowledges that adherence was not assessed (Discussion section). Explicitly stating this as a limitation in the Methods section would be transparent.
    • Sun Exposure and Dietary Intake: These are known factors influencing vitamin D levels. Mentioning whether these were considered or controlled for (even if they weren't, stating this as a limitation) would be important.
    • Timing of Blood Draws: Was there any standardization regarding the time of day for blood draws, which could influence vitamin D levels?
  • Limit: The study design is clearly stated as cross-sectional, which inherently limits the ability to establish causality. The definitions of vitamin D status are clearly provided.

Results:

  • Area for Improvement:
    • Clarity in Table 1: While the table is informative, consider adding units to the median [IQR] value for "OH-25-Hidroxi-Vitamin D" (it seems to be ng/mL, consistent with the text).
    • Figure 1 Clarity: The figure title is clear. However, ensure the axes are clearly labeled with units. It might be helpful to briefly describe the visual trend observed in the text accompanying the figure.
    • P-value Reporting in Table 2: The p-values are correctly reported.
  • Limit: The results are presented clearly and are consistent with the methodology.

Discussion:

  • Area for Improvement:
    • Connecting Results to Existing Literature: While the discussion compares the prevalence of vitamin D deficiency with other studies, further elaborating on why the deficiency persists despite supplementation (drawing on potential factors like malabsorption, adherence, genetic variations mentioned later) earlier in the discussion would strengthen the argument.
    • Implications of the 2000 IU Threshold: The finding that exceeding 2000 IU daily intake was associated with achieving sufficient levels is important. The discussion could delve deeper into the implications of this finding for local supplementation guidelines and the potential need for higher initial or maintenance doses in this population.
    • Addressing the Lack of Association with Pancreatic Enzymes: The authors suggest poor adherence as a potential reason. While plausible, exploring other possibilities (e.g., enzyme formulation, timing of intake relative to vitamin D) could be considered, even if speculative.
    • Strengthening the Link to Genetic Factors: The discussion on genetic variants affecting vitamin D metabolism and response is valuable but appears towards the end. Integrating this earlier as a potential contributing factor to the persistent deficiency, even in those receiving higher doses, would create a more cohesive narrative.
    • Clinical Significance: Briefly discussing the potential clinical implications of the observed vitamin D deficiency levels (beyond bone health, as mentioned later) in this specific cohort would enhance the impact of the findings.
    • Future Research Directions: The conclusion briefly mentions the need for interventions. The discussion could expand on specific future research directions based on the study's findings and limitations (e.g., longitudinal studies assessing adherence, impact of different supplementation formulations, genetic studies).
  • Limit: The discussion acknowledges the limitations of not assessing adherence, clinical outcomes, and genetic variants within the main body.

Conclusions:

  • Area for Improvement: The conclusion is concise. It could be strengthened by briefly reiterating the key finding regarding the association with vitamin D intake above 2000 IU and the implications for refining supplementation strategies in the Mexican pediatric CF population.
  • Limit: The conclusion accurately reflects the main takeaway message of the study.

General Comments:

  • Language and Clarity: The manuscript is written in clear and concise scientific language.
  • References: The references appear to be relevant and appropriately cited. Ensure the journal name in reference 2 is complete. The formatting of the "Nutrients" citation within the text (e.g., "Nutrients 2021, 13, x FOR PEER REVIEW 2 of 7") seems like a template artifact and should be removed in the final version.
  • Ethical Considerations: The ethical approvals and informed consent procedures are clearly stated.

Author Response

Comment 1: 

Abstract:

  • Area for Improvement: While the abstract clearly states the aim and main findings, it could benefit from briefly mentioning the implications of the persistent vitamin D deficiency. For example, a sentence like "These findings underscore the need for personalized strategies to optimize vitamin D status in this vulnerable population" could be added.
  • Limit: The abstract accurately reflects the cross-sectional nature of the study.

Answer 1: We agree with this valuable suggestion. We have mentioned some implications of vitamin D deficiency and added the suggested sentence by the reviewer to emphasize the clinical importance of new strategies to optimize vitamin D status.  

Comment 2: 

Introduction:

  • Area for Improvement: The introduction provides adequate background on CF and vitamin D deficiency. However, it could be strengthened by:
    • Quantifying the impact of vitamin D deficiency in CF: Briefly mentioning the specific consequences of vitamin D deficiency in CF patients (e.g., bone health, inflammation, pulmonary function) would emphasize the importance of the research question.
    • Elaborating on existing supplementation guidelines: Briefly mentioning the challenges or inconsistencies in current guidelines could further justify the study's aim to understand the local context.
  • Limit: The introduction appropriately highlights the lack of data in the Mexican pediatric CF population.

Answer 2. : We have expanded the introduction to mention the key consequences of vitamin D deficiency in cystic fibrosis, including impaired bone health, inflammation, and pulmonary function. Also, we briefly mentioned the inconsistencies in current CF vitamin D supplementation guidelines

Materials and Methods:

Comment 3: 

  • Area for Improvement:
    • Sample Size Justification: While 48 patients provide a snapshot, a brief justification of the sample size in relation to the prevalence of CF in the region or feasibility constraints would be beneficial.

Answer 3: The CF prevalence in Mexico is estimated by the National CF Foundation in 1/85000, but there is no national registry. We are a reference center that receives patients without social security from all over the Country and manage today a total of 50 patients. We therefore aimed to sample all the population, but 2/50 did not authorize the participation. A sample size calculation based on an expected prevalence of vitamin D deficiency of 60%, a 95% confidence level, and a precision of ±10% yielded an estimated sample size of 92. However, after applying a finite population correction for our reference center’s total CF pediatric population (50 patients), the adjusted required sample size was 33. Therefore, our final sample of 48 patients exceeded this minimum, supporting the validity of the estimates reported.

Comment 4:

    • Vitamin D Supplementation Details: While the type (D2 or D3) is mentioned, providing more detail on the form of supplementation (e.g., liquid, tablets, within multivitamins) could be relevant, as absorption might differ.

Answer 4: We added the type of formulation provided, which was cholecalciferol in tablets (chewable or dispersible). None of the patients received a liquid formulation. Some patients received a multivitamin tablet that contains ergocalciferol; eleven received only this formulation, and 29 received it in combination with cholecalciferol in tablets.

Comment 5: 

    • Adherence to Supplementation: The manuscript acknowledges that adherence was not assessed (Discussion section). Explicitly stating this as a limitation in the Methods section would be transparent.

Answer 5: Adherence considerations were incorporated into the methodology. We did not perform a specific methodological assessment to measure adherence, but we did collect data from medical records. Since no specific method was employed, the assessment of adherence was categorized as “not performed” in the first draft. However, following the reviewer's observation, we reviewed the data carefully to address this point

Comment 6: 

    • Sun Exposure and Dietary Intake: These are known factors influencing vitamin D levels. Mentioning whether these were considered or controlled for (even if they weren't, stating this as a limitation) would be important.

Answer 6: Sun exposure and dietary intake were not measured; these considerations have been incorporated as limitations of the study

Comment 7: 

    • Timing of Blood Draws: Was there any standardization regarding the time of day for blood draws, which could influence vitamin D levels?

Answer 7: All blood samples were taken early in the morning with at least 8 h of fasting. That information was added in the material and methods section.

Results:

Comment 8:

  • Area for Improvement:
    • Clarity in Table 1: While the table is informative, consider adding units to the median [IQR] value for "OH-25-Hidroxi-Vitamin D" (it seems to be ng/mL, consistent with the text).

Answer 8: IQR values were added, in addition to the units of measurement for OH-25 Hydroxy-Vitamin D.

Comment 9: 

    • Figure 1 Clarity: The figure title is clear. However, ensure the axes are clearly labeled with units. It might be helpful to briefly describe the visual trend observed in the text accompanying the figure.

Answer 9: Figure 1 was adjusted for clarity and a brief description in the figure caption.

Discussion:

Comment 10: 

  • Area for Improvement:
    • Connecting Results to Existing Literature: While the discussion compares the prevalence of vitamin D deficiency with other studies, further elaborating on why the deficiency persists despite supplementation (drawing on potential factors like malabsorption, adherence, genetic variations mentioned later) earlier in the discussion would strengthen the argument.

Answer 10: We have restructured the discussion to further elaborate on potential reasons for the persistence of vitamin D deficiency despite supplementation. Specifically, we have highlighted factors such as malabsorption, adherence to treatment, and genetic variations in vitamin D metabolism, all of which may contribute to this ongoing issue.

Comment 11: 

    • Implications of the 2000 IU Threshold: The finding that exceeding 2000 IU daily intake was associated with achieving sufficient levels is important. The discussion could delve deeper into the implications of this finding for local supplementation guidelines and the potential need for higher initial or maintenance doses in this population.

Answer 11: We have expanded the discussion on the findings related to the 2000 IU daily intake, including the implications of these results for local supplementation guidelines. We also consider the potential need for higher initial or maintenance doses in this population, based on the observed outcomes.

Comment 12: 

    • Addressing the Lack of Association with Pancreatic Enzymes: The authors suggest poor adherence as a potential reason. While plausible, exploring other possibilities (e.g., enzyme formulation, timing of intake relative to vitamin D) could be considered, even if speculative.

Answer 12: We have explored additional possibilities, such as enzyme formulation and the timing of enzyme intake in relation to vitamin D supplementation. These factors, along with adherence, are now discussed as potential contributors to the lack of significant association observed in our study.

Comment 13: 

    • Strengthening the Link to Genetic Factors: The discussion on genetic variants affecting vitamin D metabolism and response is valuable but appears towards the end. Integrating this earlier as a potential contributing factor to the persistent deficiency, even in those receiving higher doses, would create a more cohesive narrative.

Answer 13: We have integrated the discussion on genetic factors affecting vitamin D metabolism earlier in the manuscript. This section now emphasizes how genetic variations could contribute to the persistence of vitamin D deficiency, even in patients receiving higher doses of supplementation.

Comment 14: 

    • Clinical Significance: Briefly discussing the potential clinical implications of the observed vitamin D deficiency levels (beyond bone health, as mentioned later) in this specific cohort would enhance the impact of the findings.

Answer 14: We have expanded on the clinical significance of the observed vitamin D deficiency levels, particularly beyond bone health. The discussion now includes the potential implications of vitamin D deficiency on other aspects of health, such as inflammatory responses, and how these could impact clinical management.

Comment 15: 

    • Future Research Directions: The conclusion briefly mentions the need for interventions. The discussion could expand on specific future research directions based on the study's findings and limitations (e.g., longitudinal studies assessing adherence, impact of different supplementation formulations, genetic studies).

Answer 15: We have extended the conclusion to include more specific future research directions based on the study's findings and limitations. These include longitudinal studies to assess adherence, controlled trials evaluating different supplementation formulations, and genetic studies to further investigate individual responses to supplementation.

Comment 16: 

Conclusions:

  • Area for Improvement: The conclusion is concise. It could be strengthened by briefly reiterating the key finding regarding the association with vitamin D intake above 2000 IU and the implications for refining supplementation strategies in the Mexican pediatric CF population.
  • Limit:The conclusion accurately reflects the main takeaway message of the study.

Answer 16: We appreciate the reviewer’s thoughtful suggestion. In response, we have revised the conclusion to include a summary of the key finding regarding the association between higher daily vitamin D intake (>2000 IU) and the achievement of sufficient serum levels. We also emphasize the potential relevance of our findings for refining current supplementation strategies in the Mexican pediatric CF population.

Comment 17: 

General Comments:

  • Language and Clarity: The manuscript is written in clear and concise scientific language.
  • References: The references appear to be relevant and appropriately cited. Ensure the journal name in reference 2 is complete. The formatting of the "Nutrients" citation within the text (e.g., "Nutrients 2021, 13, x FOR PEER REVIEW 2 of 7") seems like a template artifact and should be removed in the final version.
  • Ethical Considerations: The ethical approvals and informed consent procedures are clearly stated.

Answer 17:

References: Corrected

Reviewer 2 Report

Comments and Suggestions for Authors

Why did the authors choose to use a cross-sectional design, instead of interventional or longitudinal study, following the dynamic pattern of vitamin D metabolism and supplement response in the CF patients?
How do the authors support the clinical significance of their findings in spite of the low number of subjects (N=48) and single-center study design that detracts from external validity?

Was there a power calculation performed to determine if the sample size was adequate to detect statistically and clinically significant differences? If not, how confident are the reported associations?
Given that levels of vitamin D are fluctuant with sun exposure and seasonality, why weren't these confounding variables supposedly controlled or measured at all for in the research?

What criteria were used to verify patient adherence to vitamin D supplementation regimens, and how do the authors explain their conclusions without this crucial information?
Why did the study not assess vitamin D binding protein levels or VDR gene polymorphisms, despite acknowledging in the discussion that these genetic factors significantly influence supplementation response?

Why do the authors describe the non-significance of association between vitamin D levels and pancreatic enzyme supplementation, not controlling for enzyme intake frequency and timing relative to meals?
Can conclusions about the efficacy of ≥2000 IU/day supplementation be made with confidence without knowing how long the supplementation lasted or adjusting for baseline vitamin D status?

Why did the study not include clinical endpoints such as lung function (e.g., FEV1), inflammatory markers, or infection rates, which would directly link vitamin D levels to CF outcomes?
How do the authors reconcile their findings with the discrepancy between CF-specific supplement use (very low in the sample) and current guideline recommendations?
Would it have been useful to stratify findings by supplement type?
Given the variation in national health care and access to supplementation, how generalizable are these findings to other CF populations globally?
If budget limitations preclude regular testing for vitamin D, as the study states, how do the authors recommend that clinicians in these environments provide monitoring and individualized care?

Author Response

Comment 1:

  1. Why did the authors choose to use a cross-sectional design, instead of interventional or longitudinal study, following the dynamic pattern of vitamin D metabolism and supplement response in the CF patients?

Answer 1: We thank the reviewer for this insightful comment. Our main objective was to evaluate the prevalence of vitamin D deficiency and its association with supplementation practices in a real-world clinical setting. With these findings we are conducting a prospective study that has been registered in clinical trials.

Comment 2:

  1. How do the authors support the clinical significance of their findings in spite of the low number of subjects (N=48) and single-center study design that detracts from external validity?

Answer 2: Thank you for this important observation. While the sample size is limited, it represents a complete sampling of the pediatric CF population under follow-up at our center. Due to the low prevalence of CF in Mexico, this approach was selected to ensure adequate representation. To clarify this, we have added a statement in the Materials and Methods section (statistical analysis paragraph) indicating that a complete sampling strategy was used. Despite the single-center design, our findings are consistent with those reported in international studies, supporting their clinical relevance. These limitations and the need for larger multicenter studies are now explicitly discussed in the revised manuscript.

Comment 3:

  1. Was there a power calculation performed to determine if the sample size was adequate to detect statistically and clinically significant differences? If not, how confident are the reported associations?

Answer 3: No formal power calculation was conducted prior to data collection, as this was an exploratory, cross-sectional analysis based on a complete sampling of all pediatric CF patients available at our center. Given the rarity of CF in Mexico, the study aimed to describe patterns of vitamin D status in this specific population rather than to test a predefined hypothesis. We acknowledge that the small sample size limits the statistical power to detect certain associations; thus, our findings should be interpreted with caution and viewed as hypothesis-generating. We have added a statement to the Limitations section of the Discussion to address this point more explicitly.

Comment 4:

  1. Given that levels of vitamin D are fluctuant with sun exposure and seasonality, why weren't these confounding variables supposedly controlled or measured at all for in the research?

Answer 4: We acknowledge that factors such as sun exposure, seasonal variation, and dietary intake are known to influence vitamin D levels and represent relevant potential confounders. Unfortunately, due to the retrospective nature of the data collection and the limited availability of standardized information in the clinical records, these variables were not formally documented or controlled in this study. This limitation has now been explicitly acknowledged and discussed in the revised manuscript (Discussion section, paragraph on limitations), highlighting the need for future prospective studies that can comprehensively assess these environmental and lifestyle factors.

Comment 5:

  1. What criteria were used to verify patient adherence to vitamin D supplementation regimens, and how do the authors explain their conclusions without this crucial information?

Answer 5: While a specific methodological tool to measure adherence (such as pill counts or pharmacy refill records) was not employed, data regarding adherence was collected from the patients’ medical records, which are based on regular clinical interviews conducted by the CF care team. This clarification has now been included in the Materials and Methods section. Given the limitations of this approach, we acknowledge that adherence could not be reliably assessed and may have influenced the observed variability in vitamin D status. We have also addressed this point in the Discussion as a study limitation. Future studies incorporating validated adherence tools are warranted to better evaluate the impact of supplementation regimens.

Comment 6:

  1. Why did the study not assess vitamin D binding protein levels or VDR gene polymorphisms, despite acknowledging in the discussion that these genetic factors significantly influence supplementation response?

Answer 6: We acknowledge in the discussion that genetic factors such as vitamin D binding protein concentrations and polymorphisms in the VDR gene can influence individual responses to supplementation. However, the primary objective of our study was to evaluate the prevalence of vitamin D deficiency and its association with clinical variables in a real-world, resource-limited clinical setting. At the time of data collection, genetic testing and measurement of vitamin D binding protein levels were not routinely available at our center, and thus could not be incorporated into the study protocol. Nevertheless, we agree that the integration of genetic and molecular data would significantly enrich the understanding of interindividual variability in vitamin D status. This has been explicitly acknowledged in the limitations and future research sections of the revised manuscript, and we consider it a valuable avenue for future investigation.

Comment 7:

  1. Why do the authors describe the non-significance of association between vitamin D levels and pancreatic enzyme supplementation, not controlling for enzyme intake frequency and timing relative to meals?

Answer 7: Fecal elastase is only measured at CF diagnosis, and it is a surrogate of exocrine pancreatic insufficiency, and those who have <200 receive enzymatic supplementation. In our study, no significant association was found between pancreatic enzyme doses and vitamin D levels. This may be due to several factors, including the timing of vitamin D intake relative to enzyme administration and individual differences in gastrointestinal absorption. Although we did not control these variables, routine clinical practice in our center recommends taking vitamin D supplements with meals and pancreatic enzymes to improve its absorption. Future studies should consider these factors to better understand the relationship between pancreatic enzyme therapy and vitamin D status in PwCF. We have clarified these points and discussed alternative explanations for the lack of association in the revised manuscript.

Comment 8:

  1. Can conclusions about the efficacy of ≥2000 IU/day supplementation be made with confidence without knowing how long the supplementation lasted or adjusting for baseline vitamin D status?

Answer 8: Thank you for your valuable comment. In response, we have added the duration of vitamin D supplementation to the results section, with a median of 4 months and an interquartile range of 3–6.75 months. An analysis was performed (though not shown in the results), which evaluated the impact of supplementation duration on vitamin D levels. However, no significant differences were observed when categorizing vitamin D status or levels based on supplementation duration. While this suggests that duration may not play a pivotal role in achieving sufficient vitamin D levels, further investigation into the interplay between baseline levels, supplementation duration, and the required daily dose would be beneficial in refining supplementation guidelines.

Comment 9:

  1. Why did the study not include clinical endpoints such as lung function (e.g., FEV1), inflammatory markers, or infection rates, which would directly link vitamin D levels to CF outcomes?

Answer 9: We agree that clinical endpoints such as lung function, inflammatory markers, and infection rates are important and could provide valuable insights into the broader implications of vitamin D status in CF, the primary objective of our study was to assess vitamin D status in this population. These clinical outcomes were not the focus of our investigation, which is why they were not included. Nevertheless, we acknowledge their relevance and have discussed them in the discussion section. Future studies that incorporate these clinical variables could provide a more comprehensive understanding of the relationship between vitamin D status and CF-related outcomes.

Comment 10:

  1. How do the authors reconcile their findings with the discrepancy between CF-specific supplement use (very low in the sample) and current guideline recommendations? Would it have been useful to stratify findings by supplement type?

Answer 10: Thank you for your comment. We have added a specification in the results section regarding the type of supplements used, noting that only 4 out of 48 patients used CF-specific supplements. Due to the lack of availability of CF-specific products in Mexico, most patients used general vitamin D supplements. While stratifying by supplement type could have been useful, the low number of patients using CF-specific supplements made this analysis unfeasible.

In the discussion, we addressed the implications of using non-CF-specific supplements and the broad recommendations in current guidelines, highlighting the need for further research on the impact of CF-specific versus general supplements in this population.

Comment 11:

  1. Given the variation in national health care and access to supplementation, how generalizable are these findings to other CF populations globally?

Answer 11: The center where this study was conducted provides care to patients who lack private or governmental insurance. Although medications are fully covered, access to them depends on administrative management. This situation is common in developing countries, where access to specialized treatments may vary, limiting the generalizability of our findings to other CF populations globally. However, the vitamin D dosing strategies suggested in our study can be adapted to the specific context of individual patients, which could strengthen the applicability of the findings to broader populations.

Comment 12:

  1. If budget limitations preclude regular testing for vitamin D, as the study states, how do the authors recommend that clinicians in these environments provide monitoring and individualized care?

Answer 12: Due to budget constraints, routine vitamin D testing is not always feasible in our center. However, we did include the duration of supplementation in the results. As stated in the revised discussion, in our center, where CF-specific supplements are not available, patients are routinely advised to take vitamin D supplements with meals and pancreatic enzymes to enhance absorption. Also, we consider that a minimal VitD dose of 2000UI will reduce the number of patients with low serum levels. This approach, while not as precise as routine testing, can still guide individualized care and help optimize patient management.

Round 2

Reviewer 2 Report

Comments and Suggestions for Authors

The cross-sectional design, small sample size (N=48), and lack of control for key confounders (sun exposure, baseline vitamin D, adherence) make it hard to draw robust, generalizable conclusions. No lung function measures, inflammatory markers, or infection outcomes were assessed, which limits the clinical implications of the vitamin D data. Without statistical power, the findings are descriptive only.

Author Response

Comment:

The cross-sectional design, small sample size (N=48), and lack of control for key confounders (sun exposure, baseline vitamin D, adherence) make it hard to draw robust, generalizable conclusions. No lung function measures, inflammatory markers, or infection outcomes were assessed, which limits the clinical implications of the vitamin D data. Without statistical power, the findings are descriptive only. 

Response: 

While the sample size is limited, it represents a complete sampling of the pediatric CF population under follow-up at our center. Due to the low prevalence of CF in Mexico, this approach was selected to ensure adequate representation.

Despite the single-center design, our findings are consistent with those reported in international studies, supporting their clinical relevance. These limitations and the need for larger multicenter studies are now explicitly discussed in the revised manuscript

The CF prevalence in Mexico is estimated by the National CF Foundation in 1/85000, but there is no national registry. We are a reference center that receives patients without social security from all over the Country and manage today a total of 50 patients. We therefore aimed to sample all the population, but 2/50 did not authorize the participation. A sample size calculation based on an expected prevalence of vitamin D deficiency of 60%, a 95% confidence level, and a precision of ±10% yielded an estimated sample size of 92. However, after applying a finite population correction for our reference center’s total CF pediatric population (50 patients), the adjusted required sample size was 33. Therefore, our final sample of 48 patients exceeded this minimum, supporting the validity of the estimates reported.